# Independent Validation of a Machine Learning Classifier for Predicting Mediastinal Lymph Node Metastases in Non-Small Cell Lung Cancer Using Routinely Obtainable [^18^F]FDG-PET/CT Parameters

**DOI:** 10.3390/curroncol32120679

**Published:** 2025-12-01

**Authors:** Agata Wdowiak, Julian M. M. Rogasch, Georg L. Baumgärtner, Nikolaj Frost, Jens-Carsten Rückert, Jens Neudecker, Sebastian Ochsenreither, Manuela Gerhold, Bernd Schmidt, Mareike Graff, Holger Amthauer, Tobias Penzkofer, Christian Furth

**Affiliations:** 1Department of Nuclear Medicine, Charité—Universitätsmedizin Berlin, Corporate Member of Freie Universität Berlin and Humboldt-Universität zu Berlin, 13353 Berlin, Germany; agata.wdowiak@charite.de (A.W.); holger.amthauer@charite.de (H.A.); christian.furth@charite.de (C.F.); 2Institute of Diagnostic and Interventional Radiology, Kantonsspital Aarau, 5001 Aarau, Switzerland; 3Berlin Institute of Health, Charité—Universitätsmedizin Berlin, 10117 Berlin, Germany; 4Department of Radiology, Charité—Universitätsmedizin Berlin, Corporate Member of Freie Universität Berlin and Humboldt-Universität zu Berlin, 13353 Berlin, Germany; georg.baumgaertner@charite.de (G.L.B.); tobias.penzkofer@charite.de (T.P.); 5Department of Infectious Diseases and Pulmonary Medicine, Charité—Universitätsmedizin Berlin, Corporate Member of Freie Universität Berlin and Humboldt-Universität zu Berlin, 13353 Berlin, Germany; nikolaj.frost@charite.de; 6Department of General, Visceral, Vascular and Thoracic Surgery, Charité—Universitätsmedizin Berlin, Corporate Member of Freie Universität Berlin and Humboldt-Universität zu Berlin, 13353 Berlin, Germany; jens-c.rueckert@charite.de (J.-C.R.); jens.neudecker@charite.de (J.N.); 7Department of Hematology and Medical Oncology, Charité—Universitätsmedizin Berlin, Corporate Member of Freie Universität Berlin and Humboldt-Universität zu Berlin, 12203 Berlin, Germany; sebastian.ochsenreither@charite.de; 8Charité Comprehensive Cancer Center, Charité—Universitätsmedizin Berlin, Corporate Member of Freie Universität Berlin and Humboldt-Universität zu Berlin, 10117 Berlin, Germany; 9Institute of Pathology, Charité—Universitätsmedizin Berlin, Corporate Member of Freie Universität Berlin and Humboldt-Universität zu Berlin, 10117 Berlin, Germany; manuela.gerhold@charite.de; 10Department of Internal Medicine—Pneumology and Sleep Medicine, DRK Kliniken Berlin Mitte, 13359 Berlin, Germany; b.schmidt@drk-kliniken-berlin.de; 11Department of Thoracic Surgery, DRK Kliniken Berlin Mitte, 13359 Berlin, Germany; m.graff@drk-kliniken-berlin.de

**Keywords:** non-small cell lung cancer, FDG-PET/CT, machine learning, lymph node staging, TCIA, NSCLC Radiogenomics

## Abstract

This study tested a previously developed machine learning model that uses routine [^18^F]FDG-PET/CT scans and clinical data to judge whether lung cancer has spread to central chest lymph nodes before treatment. Researchers applied the model, without changes, to two separate patient groups: 87 patients from one hospital and 124 patients who had primary surgery from a public dataset. They compared the model with a common PET/CT rule based on lymph node activity and size, using surgical tissue results as the standard. The rate of advanced lymph node disease differed between groups. In the hospital group, accuracy for ruling out disease was similar between methods. In the surgical group, the model produced fewer false positives than the common rule for PET/CT image interpretation. Ability to detect disease was comparable in both groups. Overall, the machine learning model’s performance was confirmed, with potential advantages in reducing false positives.

## 1. Introduction

Positron emission tomography/computed tomography (PET/CT) with [^18^F]fluorodeoxyglucose ([^18^F]FDG) is the guideline-recommended standard for pretherapeutic staging in non-small cell lung cancer (NSCLC), particularly for lymph node (LN) assessment, which is crucial for treatment planning [1]. While patients with N0/N1 status (i.e., no LN metastases or ipsilateral hilar LNs) are typically candidates for primary surgery, those with N2/N3 disease (i.e., mediastinal or contralateral hilar LN metastases) often require a non-surgical or multimodal treatment approach. However, the diagnostic accuracy of [^18^F]FDG-PET/CT for LN staging is limited by false-positive and false-negative findings [2], which can significantly impact clinical decision-making.

The primary limitation arises from the non-tumor-specific nature of [^18^F]FDG, which is taken up not only by malignant cells but also by various non-malignant, glucose-metabolizing cell types such as macrophages [3,4]. Consequently, inflammatory and benign conditions such as anthracosis, pulmonary emphysema, and epithelioid cell reactions can lead to false-positive results. Given that positive PET findings in LNs typically necessitate invasive confirmation, these inaccuracies may result in unnecessary diagnostic procedures, exposing patients to additional risks [1,5].

To address these challenges, we previously developed a machine-learning-based predictive model aimed at improving diagnostic accuracy of [^18^F]FDG-PET/CT for mediastinal LN staging in NSCLC. We placed special emphasis on increasing the specificity to reduce the need for invasive diagnostics. In 2023, we introduced a gradient boosting model incorporating 10 routinely obtainable parameters from [^18^F]FDG-PET/CT, which demonstrated a significant improvement in specificity for mediastinal LN staging compared to standard visual interpretation (73% vs. 54%) at similarly high sensitivity of approx. 90% [6]. The machine learning classifier was based on a combination of PET and CT features, including uncorrected and background-corrected maximum standardized uptake values (SUVmax) of N1 and N2 LNs, a visual PET score to rate the intensity of [^18^F]FDG uptake of mediastinal LNs, the primary tumor diameter, short axis diameters of N1 and N2 LNs, and patient age [6]. Diagnostic accuracy of the classifier was similar with two different PET/CT scanners from real-world cohorts the same institution; however, prospective and external validation remained pending.

The objective of the current analysis was to validate the model in two additional independent datasets to further assess its generalizability. This analysis is a Type 4 prediction model study according to the TRIPOD statement (transparent reporting of a multivariable prediction model for individual prognosis or diagnosis) [7]. Compliance of the methodology and reporting of the current work with established checklists is detailed in Appendix A.

## 2. Materials and Methods

### 2.1. Patients

Figure 1 illustrates the patient selection process for this analysis, which involved two distinct cohorts.

The first cohort comprised patients who were prospectively and consecutively enrolled at Charité—Universitätsmedizin Berlin (Campus Virchow-Klinikum and Campus Benjamin Franklin), a tertiary hospital, between February 2020 and October 2022 in an observational study. A positive vote from the institutional ethics committee at Charité—Universitätsmedizin Berlin was obtained (vote EA1/089/20), and all patients provided written informed consent for study participation. The study was not registered in a public trial database. These patients underwent [^18^F]FDG-PET/CT as part of routine clinical care following a recommendation from an interdisciplinary lung tumor board due to suspected lung cancer or a biopsy-confirmed initial diagnosis of lung cancer with a potentially curative treatment intent. A total of 234 patients were included in the study cohort. However, only those patients who met the following inclusion criteria were included in the current analysis: (I) Histologically confirmed diagnosis of NSCLC, (II) pre-treatment [^18^F]FDG-PET/CT fully evaluable, (III) histological data for thoracic LNs available within 6 weeks prior to and 8 weeks following the [^18^F]FDG-PET/CT, with evaluation via one of the following: (a) surgical resection with mediastinal lymphadenectomy, (b) EBUS-TBNA (endobronchial ultrasound-guided transbronchial needle aspiration) including N2/3 LNs, (c) positive result of EBUS-TBNA for N1 LNs (i.e., proven N+) and unequivocal imaging findings for mediastinal LNs. In total, 78 patients meeting these criteria for this first cohort were included in the analysis.

The second cohort comprised 124 NSCLC patients from the publicly available NSCLC Radiogenomics dataset within The Cancer Imaging Archive (TCIA) [8,9]. These patients were recruited between 7 April 2008, and 15 September 2012, at Stanford University School of Medicine and the Palo Alto Veterans Affairs Healthcare System. All patients in this cohort underwent primary surgical treatment for NSCLC. Detailed information regarding the dataset can be accessed online at [8].

As a result, 211 cases from both cohorts were ultimately included in the analysis.

### 2.2. Imaging Procotol

Patients in the first cohort (“Charité”) were examined using two different PET/CT scanners depending on whether the examination was conducted at Campus Virchow-Klinikum or Campus Benjamin Franklin. PET/CT data were acquired from the skull base to the proximal femora. Thirty-three patients from this cohort were examined with a Philips Gemini TF 16 PET/CT scanner (Koninklijke Philips N.V., Amsterdam, The Netherlands; 3D mode; transaxial field of view: 71.6 cm; acquisition time: 3 min per bed position; bed position overlap: 53.3%) [10]. This scanner uses Lutetium-Yttrium-Oxyorthosilicate (Lutetium-to-Yttrium ratio 9:1) scintillation crystals, photomultiplier tubes, time-of-flight (TOF) capability (Philips Astonish TF technology), and a system sensitivity of 6.6 cps/kBq [10]. The remaining 54 patients were scanned using a GE Discovery MI PET/CT scanner (GE Healthcare, Chicago, IL, USA; 3D mode; transaxial field of view: 70 cm; acquisition time: 2–3 min per bed position; bed position overlap: ~25%) [11]. This scanner uses Silicon Photomultipliers in a 3-ring detector arrangement, TOF capability, and a reported system sensitivity of 7.3 cps/kBq. Charité patients were required to fast for at least 6 h prior to the [^18^F]FDG injection. A median activity of 302 MBq [^18^F]FDG (IQR: 259–309 MBq; median: 3.5 MBq/kg; IQR: 3.1–4.5 MBq/kg) was administered intravenously. PET imaging was performed after a median uptake time of 64 min (IQR: 60–72 min), resulting in a decay-corrected activity of 2.3 MBq/kg at the start of image acquisition (IQR: 2.1–2.9 MBq/kg). PET data were acquired in 3D mode from the skull base to the proximal femora (transaxial field of view: 71.6 cm; acquisition time: 1.5–3 min per bed position; bed position overlap: 53.3%).

In the second cohort (“TCIA”), patients at Stanford University Medical Center were reportedly examined with a GE Discovery D690 PET/CT (GE Healthcare, Chicago, IL, USA), while Palo Alto VA used a GE Discovery PET/CT scanner. Based on information available from the DICOM header data, a median activity of 477 MBq [^18^F]FDG (IQR: 402–562 MBq; median: 6.4 MBq/kg; IQR: 5.2–7.3 MBq/kg) was administered intravenously. PET imaging was performed after a median uptake time of 66 min (IQR: 59–76 min), resulting in a decay-corrected activity of 4.1 MBq/kg at the start of image acquisition (IQR: 3.4–4.8 MBq/kg).

### 2.3. [^18^F]FDG-PET/CT Image Reconstruction

In the first cohort (“Charité”), PET raw data from the Philips Gemini TF 16 PET/CT scanner were reconstructed using an iterative reconstruction approach (Ordered Subset Expectation Maximization) with TOF analysis (BLOB-OS-TF; 3 iterations; 33 subsets; smoothing filter [kernel width: 14.1 cm; relaxation parameter: 0.7]; matrix size: 144 × 144; voxel size: 4.0 × 4.0 × 4.0 mm^3^). CT raw data were reconstructed with a slice thickness of 5 mm for attenuation correction and 3 mm for visual assessment (convolution kernel: B, optimized for soft tissue). In Charité patients examined with the GE Discovery MI PET/CT, PET raw data were reconstructed using a Bayesian penalized likelihood algorithm (GE “Q.Clear”) with a regularization parameter (β) of 450 (matrix size: 256 × 256; voxel size: 2.73 × 2.73 × 2.78 mm^3^). Non-contrast-enhanced CT raw data were reconstructed with a slice thickness of 3.75 mm for attenuation correction (convolution kernel: Q AC) and 1.25 mm for visual evaluation (convolution kernel: standard, optimized for soft tissue). Attenuation correction of all PET data in Charité patients used a native low-dose CT scan.

PET images from the TCIA cohort were generated following a similar protocol, as described by Bakr et al. [9]. At both sites, CT-based attenuation correction and iterative OSEM reconstruction were employed. However, no specific harmonization of acquisition or reconstruction protocols was conducted, as the data were collected over multiple years from different institutions.

### 2.4. Image Analysis

#### 2.4.1. SUV Measurements

SUV measurements were conducted following an identical protocol to that described in our previous publication [6], which was initially based on the methodology detailed by Toney et al. [12]. Briefly, the highest SUVmax (corrected for total body mass) within each LN region (N1, N2, N3) was determined using a three-dimensional volume of interest within the Visage 7.1 viewer (Visage Imaging, Inc., San Diego, CA, USA). Pulmonary and mediastinal background SUVmean values were determined according to the method outlined in [6] and were used to calculate LN SUVmax ratios. If a LN region was classified as PET-negative (i.e., LN uptake ≤ mediastinal background; visual score = 1), the corresponding background SUVmean was used instead of calculating an SUV ratio. Partial volume effects on the primary tumor SUVmax were corrected by applying a size-dependent recovery coefficient, as previously detailed in [6].

#### 2.4.2. Visual PET Assessment

A 4-stage score described in our previous publication [6] was used to assess the intensity of [^18^F]FDG uptake in the LNs in relation to physiological uptake of the mediastinal background or the liver. Visual assessment was performed by an experienced nuclear medicine physician (JMMR, approximately 14 years of experience in [^18^F]FDG-PET/CT for lung cancer) using the Visage 7.1 viewer. The reader was completely blinded to the results of the reference standard.

#### 2.4.3. Short-Axis Measurement of Nodal Regions by CT

The diameters of the LN short axis and the primary tumor (defined as the largest tumor diameter in the transverse plane in the lung window) were measured using contrast-enhanced CT data that were available in all cases. LN short-axis diameters were measured for each of the regions N1, N2 and N3. In each region, the LN with the largest short-axis diameter was measured, regardless of its [^18^F]FDG accumulation.

#### 2.4.4. Standard PET/CT Criterion

A LN was visually classified as positive if its mediastinal uptake exceeded the background mediastinal uptake and/or if its short-axis diameter was greater than 10 mm.

#### 2.4.5. Clinical Data Collection

Clinical and pathological data for the Charité cohort were extracted from the hospital information system and the tumor documentation system. This included demographic information (age, gender), tumor characteristics (affected side and lobe, TNM stage), and histological features (NSCLC subtype, degree of differentiation). The same data for the TCIA cohort were available as open-access data.

### 2.5. Reference Standard

The reference standard for assessing the performance of the machine learning model was the final N status, as determined by the multidisciplinary tumor board. Among the 211 patients included in the analysis, 176 patients (including all TCIA patients) underwent surgery with systematic resection of the hilar and mediastinal LNs. The remaining 35 Charité patients did not undergo surgery. In 29 of these cases, the N status was confirmed through EBUS-TBNA of the N2/3 LNs, typically using punch biopsy cylinders for histological examination. For the remaining 6 patients, no histological confirmation of N2/3 LN involvement was obtained, though EBUS-TBNA was performed on the N1 station. These patients had locally advanced or metastatic disease, and a biopsy to validate the unequivocally positive imaging findings of N2/3 LN involvement was considered unnecessary. Patients who did not undergo surgery were still included in the analysis, as the machine learning model was initially developed to be applicable to all NSCLC patients who undergo [^18^F]FDG-PET/CT prior to potentially curative treatment.

### 2.6. Application of the Machine Learning Classifier

The machine learning-based classifier was previously trained to distinguish between N0/1 and N2/3 cases. All details regarding the model development process, the predictor variables tested during development, as well as the technical aspects of the model structure and parameters of the machine learning classifier, can be found in our previous publication [6]. Briefly, the machine learning model relies on a combination of 10 parameters that include LN SUVmax, the visual PET score, primary tumor and LN diameters as well as patient age. The model consists of a pipeline implemented in Python (version 3.9) programming language that combines imputation of missing values, one-hot encoding of categorical variables, robust scaling of continuous features, machine learning-based feature selection and classification. Feature selection and final classification into “positive” (N2/N3) vs. “negative” (N0/N1) cases uses a gradient boosting model (GBM) from the sklearn library in Python. The machine learning classifier can be applied to patient data via an openly accessible web application (https://baumgagl.github.io/PET_LN_calculator/ (accessed on 18 October 2025)).

The initially trained machine-learning classifier was applied without modifications to the current dataset. As per our prior publication, a patient was classified as “positive” if the probability for N2/N3 according to the classifier exceeded 0.19 [6].

The current analysis reaches a radiomics quality score (RQS) of 24 out of 36 (67%; please refer to Appendix A for details) [13].

### 2.7. Statistical Analysis

Data collection and analysis were conducted using SPSS 28 (IBM, Armonk, NY, USA). As the Shapiro–Wilk test indicated a non-parametric data distribution, descriptive statistics were summarized using median values and interquartile ranges (IQR). For group comparisons presented in Table 1, variables were analyzed using either Fisher’s exact test or the Wilcoxon rank-sum test. Cross-tabulations were used to assess the distribution of diagnostic classifications, enabling the calculation of sensitivity, specificity and accuracy and comparison of classifications using McNemar’s test. Predictive accuracy of the machine learning classifier was further assessed with the Brier score. Net benefit of applying the classifier for lymph node staging compared to the standard PET/CT criterion was analyzed with decision curves using the dcurves package in Python version 3.9. Significance was generally assumed at α = 0.05. Statistical analysis did not follow a prespecified protocol.

## 3. Results

### 3.1. Reference Standard

A total of 50 of 211 patients (24%) were classified as N2/3 based on the histological reference standard. Due to differences in patient selection, Charité patients exhibited a significantly higher proportion of mediastinal lymph node metastases, with 35/87 patients (40%), compared to 15/124 patients (12%) in the TCIA cohort (*p* < 0.001; Table 1).

### 3.2. Diagnostic Performance

#### 3.2.1. Charité Cohort

Both the standard PET/CT criterion and the machine learning classifier demonstrated a high sensitivity of 97.1%. The machine learning classifier achieved a slightly but not significantly higher specificity (65%) compared to the standard PET/CT criterion (60%), resulting in a slightly higher accuracy (78% vs. 75%; each *p* > 0.5). For further details, please refer to Table 2 and Table 3. Figure 2 provides a case example.

#### 3.2.2. TCIA Cohort

Sensitivity with both the standard PET/CT criterion and the machine learning classifier was lower in this cohort, resulting from the preselection of only surgically treated patients. The machine learning classifier demonstrated significantly higher specificity (89% vs. 70%), leading to an improved overall accuracy of 82% compared to 65% for the standard PET/CT criterion. Full results are shown in Table 2 and Table 3. Table 4 shows results from other publications that have used the TCIA NSCLC Radiogenomics dataset to develop or test machine learning models to predict lymph node metastases.

### 3.3. Predictive Accuracy

Brier scores of the machine learning classifier were similar between the Charité cohort (0.11) and the TCIA cohort (0.097). Decision curve analysis (Figure 3) demonstrates a consistently higher net benefit of applying the machine learning classifier for lymph node staging compared to the standard PET/CT criterion.

## 4. Discussion

Accurate lymph node staging is key for optimal treatment planning in NSCLC, yet the diagnostic value of [^18^F]FDG-PET/CT remains limited by the tracer’s lack of tumor specificity, with false-positive findings often resulting from inflammatory conditions. The integration of machine learning models into this diagnostic process aims to overcome these limitations with the primary aim of enhancing specificity. The observed reduction in false positives, particularly in our previous publication [6] and in the surgically treated TCIA cohort analyzed here, may help avoid unnecessary invasive mediastinal staging procedures (EBUS-TBNA or mediastinoscopy). In cases where standard PET/CT criteria for mediastinal lymph node metastases raise suspicion of N2/3 disease, a low classifier-derived probability of N2/3 may identify patients in whom invasive staging prior to surgery is not required. This would be especially relevant in patients with a low likelihood of lymph node metastases, with suspicious lymph nodes that are anatomically difficult to access, or in multimorbid patients.

In this context, the current study demonstrates a successful independent and external validation of our previously developed machine learning classifier for pretherapeutic LN staging in NSCLC. This machine learning classifier relies on routinely obtainable parameters to ensure that it is simple to implement in any institution and to enhance its robustness and generalizability.

We observed discrepancy in sensitivity of the analyzed diagnostic criteria between the Charité and TCIA cohorts. Patients in the TCIA dataset had already undergone extensive preoperative diagnostics including [^18^F]FDG-PET/CT, making this cohort inherently different from the real-world, unselected Charité population undergoing routine clinical staging. Given that all TCIA patients were selected for primary surgery, implying they had been clinically staged as cN0/1, a sensitivity of 33% using the standard PET/CT criterion or 27% using the machine learning classifier indicates that this proportion of patients could have been correctly reclassified as cN2/3, despite an apparently negative diagnostic workup in routine clinical care. This approach is sometimes referred to as predicting occult lymph node metastases. Zhong et al. [15] used a residual neural network to predict occult N2 lymph node metastases in patients from the NSCLC Radiogenomics dataset that was the basis for the TCIA cohort in the present analysis. Using only the CT data without the [^18^F]FDG-PET images, Zhong et al. achieved an accuracy of 81%, which is similar to our model’s performance (accuracy: 82%). Other publications listed in Table 4 have achieved similar accuracy using different methodological approaches. However, Ju et al. [14] and Kidera et al. [16] focused on differentiating between N0 and N+, making their results not directly comparable to our objective of distinguishing N0/1 from N2/3. Tyagi et al. [17] reported an exceptionally high accuracy of 97% for N stage prediction using the NSCLC Radiogenomics dataset. However, their model’s performance may have been overestimated, as they used separate subsets of patients from the same dataset for training and testing, rather than validating a previously developed model on a completely independent dataset.

Importantly, our machine learning classifier would have nonetheless maintained a high specificity of 90%. These results underline the potential of the machine learning classifier as a complementary diagnostic method to routine visual interpretation of [^18^F]FDG-PET/CT images for LN staging of NSCLC to reduce the need for invasive staging methods and optimize patient stratification for the best suited therapeutic approach. This is underlined by the consistent net benefit observed in decision curve analysis. Furthermore, the observed difference in clinical characteristics between the two examined cohorts underscores the critical value of independent and external validation of any machine learning classifier to confirm its reliability in different contexts.

From a methodological perspective, this validation study achieved a radiomics quality score (RQS) [13] of 24/36 (67%), which is markedly higher than the average of 9.4/36 (26%) that Park et al. observed in their review of 77 publications on radiomics in oncologic studies [18]. Despite the extensive validation of the machine learning classifier, a prospective interventional trial is still required to assess its performance in a real-world clinical setting, where patient management decisions are directly influenced by its output. Such a trial would provide crucial insights into the actual impact of machine learning-assisted decision support on staging accuracy, treatment planning, and potentially patient outcomes.

Yoo et al. [19], Wang et al. [20], and Laros et al. [21] demonstrated that combining radiomic features from both lymph nodes and primary tumors with various clinical variables significantly improves the accuracy of predicting lymph node metastases in NSCLC. Notably, Laros et al. showed that their model achieved a higher area under the ROC curve compared to using lymph node SUVmax alone [21]. Similar to the present study, all three investigations reported strong performance of gradient boosting-based classifiers. Laros et al. further validated their model using an external cohort, in which the diagnostic accuracy was slightly reduced but remained high overall [21]. Radiomic features, particularly higher-order metrics derived from the gray-level matrices of PET images, as used by Yoo et al., Wang et al., and Laros et al., enable highly detailed analyses of tracer uptake at the voxel level. This theoretically allows for the detection of subtle imaging patterns that may not be visually discernible or intuitively interpretable by human observers. As a result, radiomics offer fundamentally new opportunities for differentiating between benign and metastatic lymph nodes, beyond the capabilities of standard visual assessment. However, radiomic features are inherently susceptible to distortion and invalidation due to numerous influencing factors during image acquisition, reconstruction, post-processing and segmentation of PET data [22,23,24]. In contrast, the simpler and less complex variables used in our model are likely more robust against such technical artifacts and may therefore offer better generalizability across different PET scanners, reconstruction protocols, and institutions.

Consistent with this assumption, our classifier has so far maintained stable performance across the analyzed cohorts despite being tested on data acquired with multiple PET scanners, acquisition protocols, and reconstruction methods. This likely reflects the robustness of the underlying features towards technical variations currently encountered in routine clinical imaging. This is underlined by our previous analysis with two different PET scanners in which SUV differed as expected but overall model predictions were rarely altered to a relevant degree [6]. However, the classifier has not yet been validated with long axial field of view PET scanners that will become more prevalent in the future. Furthermore, the current analysis underlines that the classifier’s advantage in accuracy compared to standard visual image interpretation depends on the selection criteria of the target cohort and the prevalence of N2/3. The classifier is only applicable to thoracic lymph node staging in treatment-naïve patients with first diagnosis of NSCLC. Its purpose is to support the physician’s assessment. Unlike radiomic features, the variables required for our model can be readily assessed during routine clinical interpretation using a standard image viewer. However, given that it is not a licensed medicinal product, the classifier should not be used for clinical care.

The current analysis has further limitations. The non-interventional design limits conclusions regarding the actual impact of the classifier on clinical decision-making and reduction in invasive diagnostic procedures, although this can be inferred theoretically given the established diagnostic workflows for lymph node staging in NSCLC. The robustness of the classifier when applied prospectively by readers from other institutions has not yet been formally evaluated, although the standardized collection of required parameters is expected to minimize variability.

## 5. Conclusions

This study demonstrates successful external validation of a previously developed machine learning classifier for LN staging in NSCLC. The innovative aspect of our approach lies in its combination of simplicity, interpretability, and robustness: unlike previously published high-dimensional radiomics or deep learning models, our classifier uses clinically intuitive clinical and [^18^F]FDG-PET/CT variables that can be readily assessed during routine imaging interpretation. This positions our model as a practical tool that bridges the gap between advanced computational methods and everyday clinical workflows. Its diagnostic performance remained robust across two independent cohorts, and the high radiomics quality score (RQS) achieved reflects the methodological rigor of the study, exceeding typical reporting standards in oncologic imaging research. Our work further contributes to the literature, as, to our knowledge, no other machine learning model developed for lymph node staging in NSCLC has undergone independent validation across a comparable number of datasets and PET/CT scanners.

The differences observed between the preselected TCIA cohort and the real-world Charité population underscore the importance of external validation prior to clinical implementation. While these results are promising, future work will focus on extending external validation to additional centers and independent external readers to more rigorously assess the classifier’s generalizability and robustness. Ultimately, following such further validation, our goal is to conduct a prospective interventional trial, which is essential to determine whether machine learning-assisted staging can reliably improve clinical decision-making and patient outcomes in routine practice.

## Figures and Tables

**Figure 1 curroncol-32-00679-f001:**
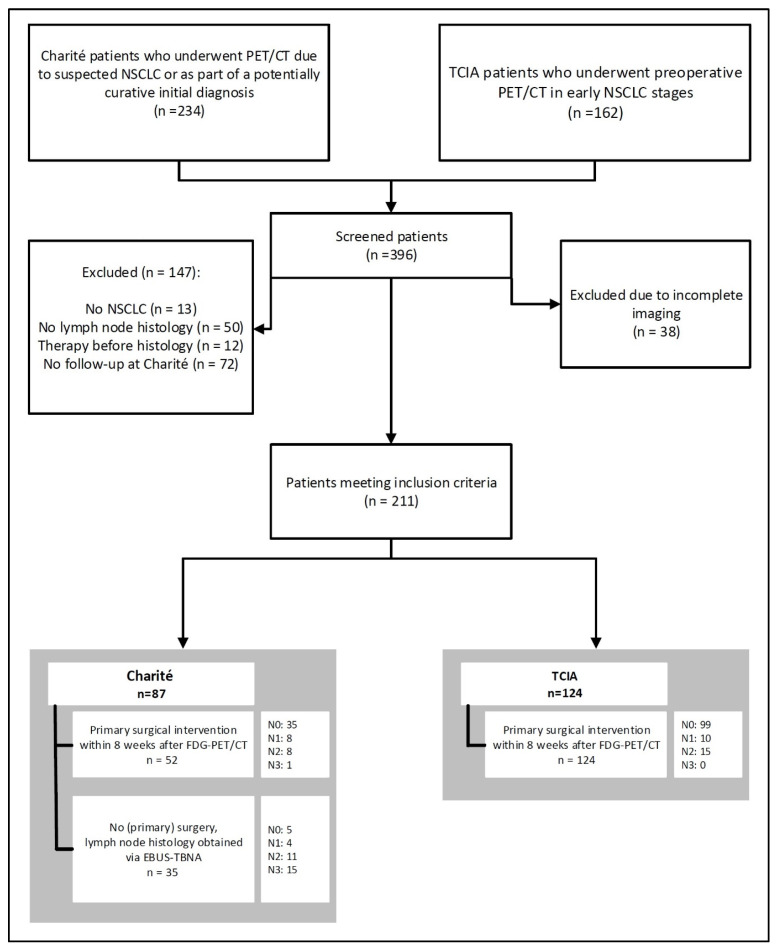
Flowchart of the patient inclusion and exclusion process.

**Figure 2 curroncol-32-00679-f002:**
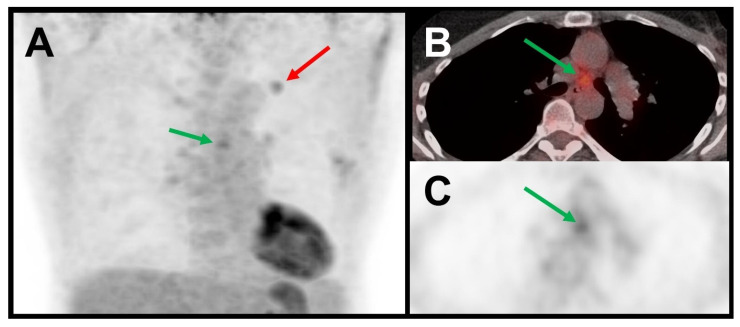
Case example. A patient with adenocarcinoma in the left upper lobe (red arrow = primary tumor) from the Charité cohort. (**A**) Maximum intensity projection; (**B**) transaxial fused PET/CT; (**C**) transaxial PET only. The patient presented with increased [^18^F]FDG uptake in mediastinal lymph node station 7 according to the International Association for the Study of Lung Cancer (IASLC) lymph node map, exceeding the background mediastinal activity (green arrows). According to the standard PET/CT criterion, this lymph node was classified as “PET/CT-positive.” Surgery with lymphadenectomy confirmed absence of lymph node metastases (pN0). The machine learning classifier predicted a probability of 0.13 for N2/N3 disease (i.e., below < 0.19), correctly classifying the patient as N0/1.

**Figure 3 curroncol-32-00679-f003:**
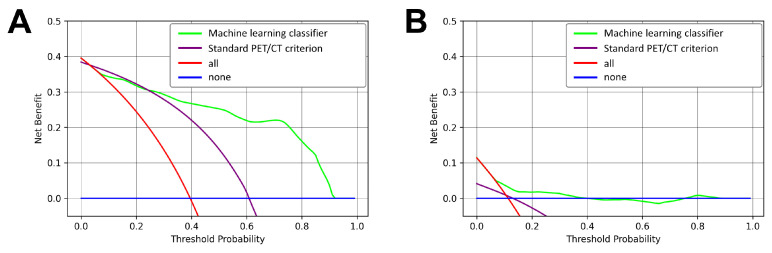
Decision curve analysis. (**A**) Charité cohort. (**B**) TCIA cohort. In both cohorts, there is a consistent net benefit of the machine learning classifier compared to the standard PET/CT criterion although the overall level of benefit differs considerably between both cohorts. The lower overall benefit of both diagnostic criteria is likely due to the strict preselection of these surgically treated patients with the standard diagnostic workup that all TCIA patients had previously undergone before being deemed candidates for primary surgical treatment.

**Table 1 curroncol-32-00679-t001:** Patient characteristics.

Parameter	Total Cohort	Charité Cohort	TCIA Cohort	*p*-Value
Number of patients	211	87	124	
Age [Years]	68 (63–74)	67 (61–72)	70 (65–76)	**0.01**
Gender: male	145 (69)	54 (62)	91 (73)	0.097
T stage				<0.001
- T1	92 (44)	29 (33)	63 (51)	
- T2	66 (31)	22 (25)	44 (36)	
- T3	32 (15)	19 (22)	13 (10)	
- T4	21 (10)	17 (20)	4 (3)	
N stage				**<0.001**
- N0	139 (66)	40 (46)	99 (80)	
- N1	22 (10)	12 (14)	10 (8)	
- N2	34 (16)	19 (22)	15 (12)	
- N3	16 (8)	16 (18)	0	
- N2/3	50 (24)	35 (40)	15 (12)	**<0.001**
Type of reference standard				**<0.001**
- surgery	176 (83)	52 (60)	124 (100)	
- EBUS-TBNA including N2/3 LNs	29 (14)	29 (33)	0	
- EBUS-TBNA of N1 LNs + definitive imaging results for N2/3	6 (3)	6 (7)	0	
Primary tumor size [mm]	34 (22–52)	31 (19–50)	38 (24–54)	0.062
Primary tumor SUVmax	7.7 (3.5–12.5)	11.1 (7.2–15)	5.4 (2.5–10.2)	**<0.001**
NSCLC subtype				0.061
- ADC	149 (70)	55 (63)	94 (76)	
- SCC	52 (25)	25 (29)	27 (22)	
- Other/NOS	10 (5)	7 (8)	3 (2)	
Primary tumor lobe				0.18
- Upper	124 (59)	46 (53)	78 (63)	
- Middle	15 (7)	5 (6)	10 (8)	
- Lower	71 (34)	35 (40)	36 (29)	
- unknown/multiple	1 (1)	1 (1)	0	
CT with contrast agent available				**<0.001**
- Yes	158 (75)	79 (91)	79 (64)	
- No	53 (25)	8 (9)	45 (36)	
Grading				**<0.001**
- G1	38 (20)	6 (7)	32 (26)	
- G2	105 (54)	34 (39)	71 (57)	
- G3	46 (24)	25 (29)	21 (17)	
- G4	4 (2)	4 (5)	0	
- unknown	18 (9)	18 (21)	0	
UICC stage				**<0.001**
- I	108 (51)	27 (30)	81 (65)	
- II	30 (14)	8 (10)	22 (18)	
- III	43 (20)	26 (30)	17 (14)	
- IV	30 (14)	26 (30)	4 (3)	

Results are presented as number (%) or median (IQR). *p*-values are derived from either the two-sided Fisher’s exact test or the Wilcoxon rank-sum test. Significant results are indicated in bold. ADC = adenocarcinoma; SCC = squamous cell carcinoma; NOS = not otherwise specified; LNs = lymph nodes.

**Table 2 curroncol-32-00679-t002:** Cross-tabulations of patient-level results.

		Reference Standard	
		N0/1	N2/3
**Charité cohort**			
Standard PET/CT criterion ^1^	N0/1	31	1
	N2/3	21	34
Machine learning classifier	N0/1	34	1
	N2/3	18	34
**TCIA cohort**			
Standard PET/CT criterion ^1^	N0/1	76	10
	N2/3	33	5
Machine learning classifier	N0/1	98	11
	N2/3	11	4

^1^ According to the standard PET/CT criterion, a lymph node was classified as positive if its mediastinal uptake exceeded the background mediastinal uptake and/or if its short-axis diameter was greater than 10 mm.

**Table 3 curroncol-32-00679-t003:** Diagnostic performance.

	Sensitivity % (95% CI)	Specificity % (95% CI)	Accuracy % (95% CI)
**Charité cohort**			
Standard PET/CT criterion	97.1 (85–100)	59.6 (45–73)	74.7 (64–83)
Machine learning classifier	97.1 (85–100)	65.4 (51–78)	78.2 (68–86)
*p*-value	1	0.5	0.55
**TCIA cohort**			
Standard PET/CT criterion	33.3 (12–62)	69.7 (60–78)	65.3 (56–74)
Machine learning classifier	26.7 (8–55)	89.9 (83–95)	82.3 (74–89)
*p*-value	1	**<0.001**	**<0.001**

95% CI = 95% confidence interval. Significant results are printed in bold. Sensitivity in the TCIA cohort was low with both criteria because all patients underwent primary surgery following preoperative staging that indicated eligibility for surgery, including [^18^F]FDG-PET/CT.

**Table 4 curroncol-32-00679-t004:** Previously published machine learning models using the NSCLC Radiogenomics dataset.

First Author	Methods	AUC	Sensitivity	Specificity	Accuracy
Ju [14]	Graph Neural Network to predict N+ vs. N0; PET + CT data	0.88	-	-	83%
Zhong [15]	Residual neural network to predict occult N2 disease; CT data only	0.81	50%	84%	81%
Kidera [16]	Convolutional neural network to predict N+ vs. N0; PET + CT data	-	73%	77%	75%
Tyagi [17]	Multi-level 3D deep convolutional neural network to predict T, N and M stage; CT data only	-	-	-	97%
Current model	GBM to predict N2/3	-	27%	90%	82%

Performance metrics are listed as reported specifically for the NSCLC Radiogenomics dataset with the exception of Kidera et al. who analyzed a mixed cohort consisting of six different datasets that included the NSCLC Radiogenomics dataset. Tyagi et al. used different subsets of patients from the dataset both for training and testing. AUC = area under the curve.

## Data Availability

All original data required for the application of the previously developed ML classifier and to reproduce the current results are openly available as anonymized patient-level data in a Zenodo repository at https://doi.org/10.5281/zenodo.17114723.

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
