# Peer review of "Independent Validation of a Machine Learning Classifier for Predicting Mediastinal Lymph Node Metastases in Non-Small Cell Lung Cancer Using Routinely Obtainable [18F]FDG-PET/CT Parameters"

_curroncol, 2025, doi:10.3390/curroncol32120679_

Round 1

Reviewer 1 Report

Comments and Suggestions for Authors

This manuscript reports an independent external validation of a previously developed machine learning classifier for predicting mediastinal lymph node involvement in NSCLC using routinely obtainable [18F]FDG-PET/CT parameters. The topic is clinically relevant, particularly because PET/CT interpretation often struggles with false positives in inflammatory or granulomatous disease. The study is well organized, and the authors provide transparent reporting with TRIPOD and RQS documentation.

The main strength of the work is the use of two external cohorts, including a real-world mixed diagnostic population and a highly selected surgical cohort. The model’s higher specificity in the surgical cohort is clinically meaningful, as reducing false positives may help avoid unnecessary invasive mediastinal staging. The data availability and fixed-threshold validation also support reproducibility.

A few points may help strengthen the manuscript:

  1. The model does not improve sensitivity but shows higher specificity in the surgical cohort. It would help to clearly state that the value lies in reducing false positives, and to discuss when this may influence decisions regarding EBUS or mediastinoscopy.
  2. Since the classifier outputs probabilities, calibration curves (and/or a Brier score) would help readers judge whether the predicted risk aligns well with observed risk across thresholds.
  3. A brief decision curve analysis or a reclassification example (e.g., “X patients might avoid invasive staging at the cost of Y missed positives”) would illustrate practical clinical impact without overextending claims.
  4. The study includes multiple scanner types and reconstruction methods. A short remark explaining why the model still performed adequately, and where variability may limit generalization, would be valuable.

These clarifications will help clinicians understand when and how the classifier may be meaningfully applied in practice.

Comments on the Quality of English Language

The English is understandable but uneven. Some sentences are long and grammar is inconsistent. Shorter and clearer sentences would improve readability. Professional editing would be beneficial.

Author Response

Please see the attachment for detailed responses. We thank the reviewer for the thorough comments to our manuscript!

Reviewer 2 Report

Comments and Suggestions for Authors
  1. Your experimental analyses are very specific. Why only the GBM method? Does GBM perform the same on different datasets? (A possibility, or not). Therefore, consider incorporating different machine learning methods (SVM, kNN, LR, etc.) into your analyses. And include the results in your paper.
  2. Your dataset is open access. Please compare/discuss with other studies in the literature that use the same dataset. Do this in the Discussion section. You may create a table.
  3. Restructure your Conclusion. Emphasize the innovative aspects of your proposed approach, its contribution to the literature, and its contributions to experts.
  4. In the Discussion section, discuss the limitations of your proposed approach.
  5. Add a final paragraph to the conclusion. Provide information about your future work.
  6. If a similar dataset is available as open source, you can perform an application analysis.

Author Response

(The authors gave the same response as above.)

Round 2

Reviewer 1 Report

Comments and Suggestions for Authors
  1. The Introduction and Methods sections now accurately frame the work as a Type 4 TRIPOD validation study, with coherent description of the two independent datasets and their differences.
  2. The revised Figure 1 and the expanded eligibility descriptions improve traceability of the sample and clarify the clinical context of each cohort.
  3. Details on PET/CT acquisition, reconstruction, feature extraction, and classifier application are now more comprehensively described. Inclusion of Brier scores and decision curve analysis strengthens the evaluation of predictive performance.
  4. The authors now handle the sensitivity discrepancy between the Charité and TCIA cohorts more carefully, and the revised Discussion appropriately explains why performance differs given the surgical preselection in TCIA.
  5. The revised manuscript demonstrates strong alignment with these frameworks, and the inclusion of full checklists in the Supplement adds transparency.
  6. Statements regarding potential clinical implications have been rewritten more cautiously, and the revised conclusion more accurately reflects the current maturity of the model.

Overall, the manuscript is now suitable for publication following minor editorial polishing.

Reviewer 2 Report

Comments and Suggestions for Authors

I requested analyses using machine learning methods. Apart from this, they contributed to my other comments. My decision: accept.